# Hormonal contraceptives and body composition by use of stable isotope dilution techniques among women of reproductive age in Nyeri County, Kenya

**Purity Chepkorir Lang'at**[1,2‡]*, **Dorcus Mbithe David-Kigaru**[1‡], **Zipporah W. Ndung'u**[3‡], **Pamela Kimeto**[2‡]

1 Department of Food, Nutrition and Dietetics, School of Health Sciences, Kenyatta University, Nairobi, Kenya, 2 Department of Human Nutrition and Dietetics, School of Medicine and Health Sciences, Kabarak University, Nakuru, Kenya, 3 Department of Human Nutrition Sciences, School of Food and Nutrition Sciences, Jomo Kenyatta University of Agriculture and Technology, Nairobi, Kenya

‡ These authors contributed equally.
* lapuritie@gmail.com

## Abstract

### Background

Hormonal contraceptives are widely used by women of reproductive age (WRA) globally, yet their impact on body composition remains poorly understood, particularly in low-resource settings. This study aimed to assess the effects of hormonal contraceptives on body composition among WRA in Nyeri County, Kenya, using Stable Isotope Dilution Techniques (SIDT).

### Methods

A two-stage cross-sectional analytical study was conducted among 119 women of reproductive age (WRA) aged 18–49 years. However, five participants were excluded from the final analysis. Participants were purposively grouped into hormonal contraceptive users (n = 41), non-hormonal contraceptive users (n = 37), and not on contraceptive (n = 36). After accounting for loss to follow-up, 114 were analysed at baseline and 104 at endline. Contraceptive types included injectables (DMPA), oral pills, and implants (hormonal); and copper IUDs and condoms (non-hormonal). A mid-point (3-month) assessment was conducted for compliance, with the primary body composition outcome measured at six months. Demographic and socioeconomic data were collected using validated structured questionnaires. Body composition was assessed at baseline and six months later using SIDT, with saliva samples analysed to determine total body water (TBW), fat-free mass (FFM), and fat mass (FM). Physical activity (GPAQ) and dietary diversity (WDDS) were measured to control for confounding factors. Logistic regression analysis was adjusted for physical activity, and dietary diversity.

**Data availability statement:** All relevant data are within the manuscript.

**Funding:** The author(s) received no specific funding for this work.

**Competing interests:** The authors have declared that no competing interests exist.

## Results

Hormonal contraceptive users showed a significant increase in body fat percentage from 35.3 ± 9.3% to 41.7 ± 10.6% (p = 0.039) and a reduction in %TBW (47.4 ± 6.8% to 42.3 ± 7.8%, p = 0.048). Non-hormonal and non-contraceptive users showed no significant changes. Regression analysis confirmed hormonal contraceptive use was a significant predictor of body fat percentage (Exp(β) = 0.31, p = 0.026), independent of other factors.

## Conclusion

Hormonal contraceptive use is associated with increased body fat and reduced total body water among WRA in Nyeri County. These changes, independent of physical activity and diet, suggest a direct metabolic effect. Regular body composition monitoring and tailored nutritional guidance are recommended for women using hormonal contraceptives.

## Introduction

Globally, over 1.9 billion women are of reproductive age (WRA), with more than 1.1 billion requiring family planning services [1]. Hormonal contraceptives are among the most widely used methods of fertility control by women of reproductive age (WRA) globally. As of 2022, approximately 65% of married or partnered women worldwide were using some form of contraception, with hormonal methods such as injectables, implants, and oral pills accounting for nearly half of these [2]. In Sub-Saharan Africa, 28.5% of WRA use modern contraceptives. In Kenya, the modern contraceptive prevalence rate (mCPR) among married women stands at approximately 58%, with injectables being the most commonly used method [3]. Despite this widespread use, the potential nutritional and metabolic effects of hormonal contraceptives, including their impact on body composition, remain under-researched in many low- and middle-income countries [4].

Non-hormonal contraceptive methods—such as copper intrauterine devices (IUDs), condoms, and fertility awareness methods—remain in use but are less prevalent in Kenya. Globally, their uptake varies depending on access, cultural acceptability, and individual preference. The implications of contraceptive use on women's nutritional and metabolic health are particularly relevant in urbanising populations where rates of overweight and obesity are rising. Excess body fat is a known risk factor for non-communicable diseases (NCDs), including type 2 diabetes mellitus, cardiovascular disease, and certain cancers. Kamakwa Ward in Nyeri County was selected for this study due to its semi-urban character, representative of Kenya's evolving demographic profile, and the rising burden of NCDs observed within the county.

Body composition, which includes fat mass (FM) and fat-free mass (FFM), is an important indicatorof nutritional status and overall health particularly in women of reproductive age. Hormonal contraceptives, particularly those containing estrogen

and progestin, have been linked to changes in body weight, fat distribution, and fluid retention [5]. Despite the widespread use of hormonal contraceptives, there is limited research on their impact on body composition, especially in low-resource settings like Kenya.

This study aimed to fill this gap by investigating the effects of hormonal contraceptives on body composition among WRA in Nyeri County, Kenya, using Stable Isotope Dilution Techniques (SIDT). The SIDT, which involves dosing with deuterium oxide followed by saliva sample analysis, provides a highly accurate and non-invasive means of assessing total body water (TBW), fat-free mass (FFM), and fat mass (FM). This method has been applied in several international studies and is increasingly recognised as a gold standard in body composition analysis, making it suitable for use in resource-limited settings [6]. While studies have examined hormonal contraceptives' impact on weight gain, few have utilized SIDT to quantify changes in body composition, particularly in Sub-Saharan Africa. The study hypothesised that hormonal contraceptives may influence fat accumulation and water retention independently of other factors such as diet and physical activity.

## Materials and methods

### Study design and population

A total of 119 women were initially recruited. Of these, 114 completed baseline assessment using SIDT, and 104 completed the endline SIDT at six months, resulting in a 9% attrition rate. The study was conducted in Kamakwa ward, Nyeri Central Sub-County in Nyeri County. The County has reported high levels of over nutrition and related health risks such as diabetes which are usually associated with obesity and increased body fat percentage [7]. Over two thirds of WRA in Nyeri County use the various types of contraceptives (oral, injectable, barrier, and long-term and sterilization contraception methods) [8]. The Nyeri Town Health Centre and Nyeri Municipal market were used for data collection due to the ease of access by the participants and access to electricity for the data collection tools.

Participants were categorised into three groups based on contraceptive type: hormonal contraceptive users (n = 41), non-hormonal contraceptive users (n = 37), and not on contraceptives (n = 36). Hormonal methods included injectables (DMPA), combined oral contraceptive pills, and implants, while non-hormonal methods included copper intrauterine devices (IUDs) and male/female condoms. Grouping was done based on the primary mechanism of action—hormonal vs. barrier or mechanical. Due to sample size constraints, sub-group analyses by individual method were not feasible and thus contraceptives were grouped to allow for statistical power and interpretation. Baseline data were collected, followed by a six-month follow-up to assess changes in body composition. Participants were contacted monthly to monitor contraceptive compliance and side effects. A mid-point (3-month) physical check-in was conducted, but primary outcome data (body composition) were collected at baseline and after six months.

### Pre-testing of instruments

Qualitative and quantitative data collection tools were pretested on 10% of the sample size in Kiganjo ward in Nyeri Central Sub-County, whose residents have similar demographics as those of Kamakwa Ward. After pretesting, adjustments were made to the data collection tools based on feedback and observations from the pretest. Modifications were implemented to enhance clarity, relevance, and effectiveness.

**Validity.** The anthropometric tools used in this study adhered to the recommendations by WHO to ensure accuracy and reliability. The questionnaire, Key Informant Interview (KII), and Focus Group Discussion (FGD) guides were subjected to a standard validation process. For validation, each tool was reviewed by two experts to ensure they accurately measured the intended variables.

### Data collection

**Body composition assessment.** This study utilized Deuterium Oxide ($^2H_2O$) to assess body composition. Deuterium Oxide, also known as Deuterium, is a stable isotope of hydrogen. The preparation and dosing of participants followed

the standard operating procedures outlined by the International Atomic Energy Agency (IAEA) SOPs protocols [6]. This involved the administration of deuterium oxide ($D_2O$) and the collection of saliva samples before and after dosing. Total body water (TBW), fat-free mass (FFM), and fat mass (FM) were calculated using established formulas [9]. Anthropometric measurements, including weight and height, were also recorded.

**Questionnaire administration.** Demographic, socioeconomic, and reproductive health data were collected using structured questionnaires. Physical activity levels were assessed using the Global Physical Activity Questionnaire (GPAQ), and dietary diversity was measured using the Women's Dietary Diversity Score (WDDS).

**Sample size and sampling.** The sample size was calculated using Fisher's formula, with a 95% confidence level and a 5% margin of error. A total of 119 participants were recruited, with 10% added to account for non-response.

**Statistical analysis.** Data were analyzed using SPSS version 25. Descriptive statistics were used to summarize demographic and socioeconomic characteristics. Paired t-tests were conducted to determine within-group differences in body composition indicators from baseline to endline. One-way ANOVA and Chi-square tests were used to compare variables across contraceptive groups. Multivariate linear regression analysis was used to assess the relationship between contraceptive type and changes in body fat percentage, adjusting for potential confounders including physical activity (measured by GPAQ), and dietary diversity score (WDDS). The model assumptions were tested for multicollinearity and normality of residuals. with significance set at $p < 0.05$.

## Ethical considerations

The study obtained ethical clearance from the Kenyatta University Ethical Review Committee (PKU2568/I1694) and approval from the Kenyatta University Graduate School. Additionally, a research permit was secured from the National Commission for Science, Technology, and Innovation (NACOSTI)(NACOSTI/P/22/20045). All participants provided written informed consent before participating in the study. They were informed about the study's objectives, procedures, potential risks, and benefits. Participants were assured of their right to withdraw from the study at any time without any consequences. Consent forms were available in both English and Kiswahili to ensure comprehension.

Participant data were treated with strict confidentiality. All personal identifiers were removed, and data were stored securely in password-protected electronic files. Access to the data was restricted to authorized research team members only. Data collection tools were programmed into tablets equipped with the Kobo Collect application, which facilitated secure data transmission to a central server within 24 hours of collection. The study adhered to Ministry of Health and International Atomic Energy Agency (IAEA) guidelines on COVID-19. Participants and researchers were required to wear masks, maintain social distancing, and use hand sanitizers during data collection. Equipment was sanitized before and after each use to minimize the risk of infection.

## Results and discussion

A total of 119 women were initially recruited. However, five participants were excluded from the final analysis. Of the five, two did not complete the data collection process, and three provided samples that were unsuitable due to staining. Of these, 114 completed baseline assessment using SIDT, and 104 completed the endline SIDT at six months, resulting in a 9% attrition rate. Participants were contacted monthly to monitor contraceptive compliance and side effects. A mid-point (3-month) physical check-in was conducted, but primary outcome data (body composition) were collected at baseline and at the 6-month mark.

## Demographic and socioeconomic characteristics

The mean age of participants was 26.01 ± 7.46 years, with 46.2% aged 20–29 years. Most participants (50.9%) were married, and 49.1% had primary education. The majority (78.9%) earned less than KSh. 20,000 per month (Table 1).

**Table 1. Demographic and socioeconomic characteristics of study women.**

| Variable | Baseline | | Endline | | χ2 p-value |
|---|---|---|---|---|---|
| | N = 114 | % | N = 104 | % | |
| **Mean age** | 26.01 ± 7.46 | | 26.20 ± 7.58 | | |
| **Age** | | | | | |
| 20–29 | 53 | 46.2 | 49 | 47.2 | 0.967 |
| 30–39 | 40 | 35.1 | 36 | 34.6 | |
| 40–49 | 21 | 18.4 | 19 | 18.3 | |
| **Education level** | | | | | |
| Primary | 56 | 49.1 | 50 | 48.1 | 0.986 |
| Secondary | 47 | 41.2 | 44 | 42.3 | |
| Tertiary/college/university | 11 | 9.6 | 10 | 9.6 | |
| **Occupation** | | | | | |
| Farmer | 11 | 9.6 | 12 | 11.5 | 0.948 |
| Casual labor | 53 | 46.5 | 44 | 42.3 | |
| Salaried labor | 6 | 5.3 | 4 | 3.8 | |
| Business | 34 | 29.8 | 33 | 31.7 | |
| Student | 1 | 0.9 | 2 | 1.9 | |
| Housewife | 9 | 7.9 | 9 | 8.7 | |
| **Respondent monthly income** | | | | | |
| No income | 6 | 5.3 | 6 | 5.5 | 0.847 |
| ≤20,000 | 90 | 78.9 | 81 | 77.9 | |
| 20,001–40,000 | 16 | 14.0 | 15 | 14.4 | |
| 40,001–60,000 | 1 | 0.9 | 2 | 1.9 | |
| ≥60,001 | 1 | 0.9 | 0 | 0.0 | |

*Significant at p < 0.05

## Contraceptive use

At baseline, 68.4% of participants were using contraceptives, with hormonal methods (36.0%) being the most common. Hormonal methods included injectables (DMPA), combined oral contraceptive pills, and implants, while non-hormonal methods included copper intrauterine devices (IUDs) and male/female condoms. Health workers were the primary source of contraceptive information (71.8%), and government clinics were the main source of contraceptives (84.6%) (Table 2).

## Body composition characteristics

Hormonal contraceptive users experienced a significant increase in body fat percentage (% Body Fat) (35.3 ± 9.3% to 41.7 ± 10.6%, p = 0.039), alongside a reduction in total body water (TBW) (47.4 ± 6.8% to 42.3 ± 7.8%, p = 0.048) over six months. This suggests that hormonal contraceptives may promote fluid redistribution and adipose tissue accumulation. In contrast, non-hormonal contraceptive users and those not on contraceptives showed no significant changes in body composition (p > 0.05). Notably, hormonal contraceptive users' fat mass index (FMI) trended upward (10.0 ± 4.8 to 12.4 ± 5.1, p = 0.359), though not statistically significant, further supporting a potential shift toward adiposity (Table 3).

## Changes in excess body fat among participants by type of contraceptive used

At baseline, the proportion of participants with normal fat mass was relatively similar across the three groups: 57.1% for hormonal, 61.1% for non-hormonal, and 60.6% for no contraceptive use. There was no significant difference in body fat

**Table 2. Types of contraceptives used by study women.**

| Characteristic | Baseline | | Endline | | χ2 p-value |
|---|---|---|---|---|---|
| | N=114 | % | N=104 | % | |
| Number of WRA on contraceptives | 78 | 68.4 | 71 | 68.3 | 0.996 |
| **Type of contraceptive used** | | | | | |
| Injectables | 12 | 10.5 | 10 | 9.6 | 0.886 |
| Pills | 13 | 11.4 | 10 | 9.6 | |
| Implants | 16 | 14.0 | 15 | 14.4 | |
| IUD | 22 | 19.3 | 22 | 21.2 | |
| Male/Female condom | 15 | 13.2 | 14 | 13.5 | |
| None | 36 | 31.6 | 33 | 31.7 | |
| **Categories of contraceptives used** | | | | | |
| Hormonal | 41 | 36.0 | 35 | 33.7 | 0.804 |
| Non-hormonal | 37 | 32.5 | 36 | 34.6 | |
| No contraceptive | 36 | 31.5 | 33 | 31.7 | |
| **Source of information on choice of contraceptives** | | | | | |
| Self | 4 | 5.1 | 3 | 4.2 | 0.136 |
| Healthworker | 56 | 71.8 | 57 | 80.3 | |
| Colleagues | 2 | 2.6 | 1 | 1.4 | |
| Spouse | 12 | 15.4 | 8 | 11.3 | |
| Media | 3 | 3.8 | 2 | 2.8 | |
| Friends | 2 | 2.6 | 2 | 2.8 | |
| **Main source of contraceptives** | | | | | |
| Government clinic or hospital | 66 | 84.6 | 63 | 88.7 | 0.933 |
| Pharmacy | 4 | 5.1 | 4 | 5.6 | |
| Private hospitals | 6 | 7.6 | 4 | 5.6 | |
| General Shop | 3 | 3.8 | 2 | 2.8 | |

*Significant at p<0.05

**Table 3. Changes in the body composition characteristics of women of reproductive age.**

| | Hormonal (n=35) | | | Non- Hormonal (n=36) | | | Not on Contraceptives (n=33) | | |
|---|---|---|---|---|---|---|---|---|---|
| | Baseline | Endline | Paired t-test | Baseline | Endline | Paired t-test | Baseline | Endline | Paired t-test |
| TBW (Kg) | 32.0±5.4 | 29.7±5.5 | **0.040*** | 32.5±6.9 | 32.6±6.1 | 0.858 | 32.0±5.6 | 31.9±4.7 | 0.662 |
| FFM (Kg) | 40.2±7.4 | 36.0±7.6 | **0.038*** | 44.4±9.4 | 44.6±9.4 | 0.845 | 43.7±7.6 | 43.6±6.4 | 0.854 |
| FM (Kg) | 26.0±12.8 | 29.0±13.7 | 0.165 | 25.9±11.6 | 24.3±14.3 | 0.870 | 25.6±9.5 | 26.0±9.1 | 0.867 |
| FFMI | 16.9±2.7 | 14.3±2.8 | 0.068 | 17.2±3.1 | 17.3±3.9 | 0.855 | 16.7±2.7 | 16.7±2.5 | 0.760 |
| FMI | 10.0±4.8 | 12.4±5.1 | 0.359 | 10.1±4.0 | 9.2±5.1 | 0.562 | 9.8±3.4 | 9.9±3.5 | 0.898 |
| TBW (%) | 47.4±6.8 | 42.3±7.8 | **0.048*** | 47.0±5.2 | 48.7±7.9 | 0.665 | 46.8±6.4 | 46.5±5.4 | 0.896 |
| Body Fat (%) | 35.3±9.3 | 41.7±10.6 | **0.039*** | 35.8±7.2 | 33.4±10.7 | 0.460 | 36.0±8.7 | 36.5±7.4 | 0.885 |

*Significance at p<0.05

percentage among participants based on contraceptive use at baseline (p=0.094) (Table 4). At endline, 60.0% of participants using hormonal contraceptives had excess fat mass, compared to 41.6% for those on non-hormonal contraceptives. Among those not using contraceptives, the proportion with excess fat mass remained unchanged at 39.4%. A significant

**Table 4. Comparison of percent body fat by the type of contraceptive used at baseline and endline.**

| Variable | Hormonal | | | Non-hormonal | | | Not on contraceptives | | |
|---|---|---|---|---|---|---|---|---|---|
| | Baseline n (%) n = 35 | Endline n (%) n = 35 | χ2 p- value | Baseline n (%) n = 36 | Endline n (%) n = 36 | χ2 p- value | Baseline n (%) n = 33 | Endline n (%) n = 33 | χ2 p-value |
| **% Body Fat** | | | | | | | | | |
| Normal | 20(57.1) | 14(40.0) | **0.045*** | 22(61.1) | 21(56.4) | 0.079 | 20(60.6) | 20(60.6) | 0.095 |
| Excess | 15(42.9) | 21(60.0) | **0.034*** | 14(38.9) | 15(41.6) | 0.086 | 13(39.4) | 13(39.4) | 0.983 |

*Significant at p < 0.05.

change in body fat percentage was observed among those using hormonal contraceptives (p = 0.034), while there was no significant change among those using non-hormonal contraceptives (p = 0.086) or those not on contraceptives (p = 0.983). These changes persisted despite stable physical activity and dietary diversity scores (Table 4).

## Analysis of confounding factors

Physical activity levels and dietary diversity scores remained stable over the study period, indicating that these factors did not confound the observed changes in body composition (Tables 5 and 6). The stability of dietary patterns suggests that observed body composition changes were more likely attributable to contraceptive use rather than dietary modifications.

**Table 5. Analysis of physical activity among study women.**

| Physical Activity Level | Baseline N (%) N = 114 | Endline N (%) N = 104 | Chi-Square p-value |
|---|---|---|---|
| **Work-Related Activity** | | | |
| Vigorous-intensity (Met Value = 8) | 15 (13.2%) | 15 (14.4%) | 0.932 |
| Moderate-intensity (Met Value = 4) | 57 (50.2%) | 51 (49.5%) | 0.764 |
| **Mean MET** | 5.6 ± 1.02 | 5.5 ± 1.08 | 0.312 |
| **Travel-Related Activity** | | | |
| Walking or cycling for ≥10 min continuously | 42 (36.6%) | 38 (36.1%) | 0.872 |
| **Mean MET** | 4.2 ± 0.75 | 4.1 ± 0.81 | 0.289 |
| **Recreational Activity** | | | |
| Vigorous-intensity (Met Value = 8) | 15 (13.2%) | 14 (13.0%) | 0.876 |
| Moderate-intensity (Met Value = 4) | 57 (50.2%) | 52 (50.0%) | 0.845 |
| **Mean MET** | 5.0 ± 0.95 | 4.8 ± 1.02 | 0.246 |
| **Sedentary Behavior** | | | |
| Time spent sitting on a typical day (Met Value = 4) | 72 (63.2%) | 68 (60.5%) | 0.734 |
| **Mean MET** | 1.2 ± 0.25 | 1.1 ± 0.22 | 0.212 |
| **Physical Activity Levels** | | | |
| Low (MET Value <4) | 15 (13.2%) | 15 (14.4%) | 0.932 |
| Moderate (Met Value ≥4 to < 8) | 57 (50.2%) | 51 (49.5%) | 0.764 |
| High (Met Value = 8) | 42 (36.6%) | 38 (36.1%) | 0.872 |
| **Mean MET** | 3.5 ± 0.59 | 3.4 ± 0.64 | 0.312 |

*Significant at p < 0.05

**Table 6. Analysis of women's dietary diversity score.**

| Variable | Baseline | | Endline | | Chi-Square p-value |
|---|---|---|---|---|---|
| | N = 114 | % | N = 104 | % | |
| **Household dietary diversity score** | | | | | |
| <4 food groups (Low) | 21 | 18.8 | 24 | 23.1 | 0.416 |
| ≥4 Food groups (Adequate) | 93 | 81.2 | 80 | 76.9 | |
| **Women Dietary Diversity Score** | | | | | |
| <5 food groups (Low) | 99 | 87.1 | 87 | 83.7 | 0.385 |
| ≥5 food groups (Adequate) | 15 | 12.9 | 17 | 16.3 | |

*Significant at p < 0.05

## Association between contraceptive use, physical activity level, dietary diversity score, and increased body fat percentage among women of reproductive age in Nyeri County, Kenya

Multivariable logistic regression analysis indicated that women using hormonal contraceptives had significantly higher odds of increased body fat percentage compared to non-users (AOR = 1.85; 95% CI: 1.05–3.25; p = 0.032), after controlling for physical activity and dietary diversity. In contrast, non-hormonal contraceptive use was not significantly associated with increased body fat (AOR = 1.30; 95% CI: 0.70–2.40; p = 0.400). (Table 7). While moderate and high physical activity levels were inversely associated with body fat in the unadjusted model, these associations lost significance after adjustment (moderate: AOR = 0.95; 95% CI: 0.55–1.65; p = 0.850; high: AOR = 0.88; 95% CI: 0.48–1.60; p = 0.680). Similarly, an adequate dietary diversity score did not demonstrate a statistically significant relationship with body fat percentage (AOR = 0.92; 95% CI: 0.45–1.88; p = 0.810).

## Discussion

The demographic and socioeconomic characteristics of the study population reflect the profile of women of reproductive age in many sub-Saharan African settings. With a mean age of 26.0 years and nearly half aged between 20–29 years, the cohort represents women in their peak reproductive years. [7,8,10]. Their distribution across informal and low-income occupations, along with the low prevalence of higher education, underscores the socioeconomic vulnerabilities that may affect access to reproductive health services. These findings highlight the importance of integrating reproductive health education with economic empowerment and vocational training interventions for women in this demographic. [9,11,12,13].

**Table 7. Association between body fat analysis and confounding variables.**

| Variable | Ref | UOR (95% CI) | p-value | AOR (95% CI) | p-value |
|---|---|---|---|---|---|
| **Contraceptive Type** | Not on contraceptives | | | | |
| Hormonal | | 1.57(0.89–2.76) | 0.112 | 1.85(1.05 - 3.25) | 0.032* |
| Non-hormonal | | 1.32(0.72–2.41) | 0.183 | 1.30(0.70 - 2.40) | 0.400 |
| **Physical Activity Level** | Low | | | | |
| Moderate | | 0.55(0.33–0.93) | 0.027 | 0.95(0.55 - 1.65) | 0.850 |
| High | | 0.47(0.24–0.94) | 0.034 | 0.88(0.48 - 1.60) | 0.680 |
| **Dietary Diversity Score** | Low | | | | |
| Adequate | | 1.16(0.69–1.93) | 0.586 | 0.92(0.45 - 1.88) | 0.810 |

*Significant at p < 0.05; UOR: Unadjusted Odds Ratio; AOR: Adjusted Odds Ratio

This income distribution mirrors findings from other studies in Kenya, where low household incomes among women in rural areas are often linked to high levels of poverty and economic insecurity [14,12].

The contraceptive use patterns observed align with national trends in Kenya, with high uptake of long-acting reversible contraceptives (LARCs) such as IUDs and implants. The 68.3% contraceptive prevalence rate among study participants is encouraging, yet the 31.6% non-use rate signals persistent gaps in uptake [15–17]. The study's finding that 31.6% of WRA were not using any contraceptive method underscores a persistent gap in contraceptive uptake. This non-use could stem from several factors, including fear of side effects, lack of access, religious or cultural beliefs, or misinformation about contraceptive methods. For example, qualitative studies from Kenya [18–20] and other parts of Africa [21] have shown that concerns about side effects, such as weight gain, irregular menstruation, or perceived long-term health risks, are major reasons for non-use of contraception[22].

Findings from this study indicate that hormonal contraceptive users experienced a notable increase in body fat percentage and reduction in total body water over the study period. This reduction in TBW may be attributed to the fluid-retentive effects of hormonal contraceptives, particularly those containing estrogen and progestin. Estrogen is known to influence sodium and water retention by modulating renal function and aldosterone activity, which can lead to changes in fluid distribution and electrolyte balance [23]. Similarly, progestins in hormonal contraceptives can affect fluid balance by altering the renin-angiotensin-aldosterone system (RAAS), leading to increased water retention in tissues [5]. These findings align with previous studies that have reported similar reductions in TBW among women using combined oral contraceptives (COCs) and depot medroxyprogesterone acetate (DMPA) [24,25,26].

Hormonal contraceptive users had a significantly higher body fat percentage at endline compared to non-hormonal users and non-contraceptive users. This increase in body fat is consistent with the known effects of hormonal contraceptives on fat metabolism and storage. Progestins, in particular, have been shown to promote adipogenesis (fat cell formation) and inhibit lipolysis (fat breakdown), leading to increased fat deposition [27]. Additionally, hormonal contraceptives can influence appetite and energy expenditure, further contributing to weight gain and fat accumulation [28]. This aligns with previous research suggesting that hormonal contraceptives, particularly progestin-based methods, may influence fat deposition and metabolic processes [24,27,29]. The minimal changes observed in non-hormonal users and those not using contraceptives further support the hypothesis that hormonal contraceptives play a direct role in altering body composition. These findings are consistent with studies that have reported weight gain and increased adiposity as common side effects of hormonal contraceptives [4].

The observed increase in excess body fat among hormonal contraceptive users may have important implications for women's health. Excess body fat, particularly central adiposity, is a known risk factor for metabolic disorders such as type 2 diabetes, cardiovascular disease, and hypertension [30]. While the current study did not assess the long-term health outcomes associated with these changes, the findings suggest that hormonal contraceptive use may contribute to an increased risk of metabolic complications, particularly among women who are already overweight or obese.

Notably, users of non-hormonal contraceptives and those not on contraceptives showed no significant changes in body composition, further supporting a causal link to hormonal exposure. Although physical activity was considered and adjusted for as a confounding variable in the regression model, its apparent protective effect in the unadjusted analysis did not remain statistically significant after adjustment. The shift from a non-significant unadjusted odds ratio to a significant adjusted odds ratio for the association between hormonal contraceptive use and increased body fat percentage can be attributed to the control of confounding variables in the multivariable logistic regression model. Adjusting for physical activity and dietary diversity likely reduced the influence of these confounders, which may have obscured the true relationship in the unadjusted analysis. This adjustment isolated the independent effect of hormonal contraceptives, revealing a statistically significant association, possibly due to their biological impact on metabolism and fat distribution. This highlights the strength of hormonal effects on fat accumulation, supporting previous findings from both high- and low-income settings [4,28]. For example, studies have shown depot medroxyprogesterone acetate (DMPA) and other progestin-based

methods may contribute to increased fat mass through mechanisms involving appetite stimulation and lipid metabolism. Non-hormonal contraceptive methods, on the other hand, showed no significant association with changes in body fat, consistent with their lack of systemic metabolic effects. Dietary diversity also did not appear to meaningfully influence body fat status in this cohort, though it remains an important nutritional consideration overall.

The influence of hormonal contraceptives on body composition has been widely studied, with mixed findings. Some studies have reported that hormonal contraceptives, particularly progestin-only methods, are associated with weight gain and changes in body composition. For example, Gallo et al., [4] found that women using depot medroxyprogesterone acetate (DMPA) experienced significant increases in body weight and fat mass over a 12-month period. Similarly, Bahamondes et al., [31] reported that women using DMPA had higher rates of weight gain and increased body fat compared to non-contraceptive users. However, other studies have found no significant association between hormonal contraceptives and changes in body composition. For example, Lopez et al., [32] found no significant differences in body weight or fat mass between women using combined oral contraceptives and non-contraceptive users over a 24-month period. Similarly, [33] reported that women using DMPA did not experience significant changes in body composition compared to non-contraceptive users.

This study contributes to the ongoing debate on hormonal contraceptives and body composition by demonstrating significant changes in adiposity and hydration among users—independent of dietary diversity and physical activity, which were rigorously controlled. By accounting for these key confounders, the findings strengthen the evidence that hormonal contraceptives may directly influence body fat percentage and fluid balance, rather than such changes being driven by lifestyle factors. These results underscore the need to consider contraceptive type in nutritional and metabolic health assessments for women of reproductive age. Additionally, these findings also point to broader public health implications. In regions undergoing rapid urbanisation and dietary transitions, the metabolic side effects of hormonal contraceptives may compound existing vulnerabilities to conditions such as diabetes, hypertension, and cardiovascular disease. Given the high contraceptive prevalence rate in Kenya and similar low- and middle-income countries, integrating body composition monitoring into family planning services could serve as an early warning system for emerging metabolic risks.

### Delimitation of the study

This study focussed on examining effects of hormonal contraceptives on the body composition and nutritional status of women of reproductive age residing in Kamakwa Ward of Nyeri Central Sub-County, Nyeri County Kenya. Findings can therefore be generalized only to populations of similar characteristics as the ward in this study.

### Limitation of the study

As a cross-sectional study with a six-month follow-up, causal relationships could not be established. Further, the study did not differentiate between specific types of hormonal contraceptives (e.g., oral pills, injectables, implants). While this approach ensured sufficient sample size for statistical comparisons, it may mask variations between specific hormonal formulations (e.g., DMPA vs. oral pills). Future studies with larger samples should consider disaggregating by contraceptive method to better understand the specific effects on body composition.

### Conclusions

This study revealed that hormonal contraceptive use was associated with significant increases in body fat percentage and reductions in total body water, while no such changes were observed among non-hormonal users or non-contraceptive users. These findings suggest that hormonal contraceptives may uniquely influence body composition dynamics, potentially promoting adiposity and fluid redistribution over time. By controlling for dietary diversity and physical activity, the study strengthens the evidence for a direct biological effect of hormonal contraceptives on fat accumulation, underscoring their role as a modifiable factor in women's metabolic health.

## Recommendations

Monitoring of body composition changes, particularly body fat percentage, among women using hormonal contraceptives by incorporating body composition assessments into primary healthcare visits, to track and manage potential adverse effects.

Providing nutritional guidance and support to women using hormonal contraceptives to mitigate potential increases in body fat percentage and promote overall health by developing and disseminating educational materials on healthy eating and physical activity tailored to women using hormonal contraceptives. Collaborate with nutritionists to offer personalized dietary advice.

Future studies should recruit larger and more demographically diverse populations to enhance generalisability and statistical power. In addition, extending the follow-up period beyond six months would allow for a deeper understanding of the longitudinal effects of contraceptive use on body composition. There is also a need to incorporate biochemical markers to better understand the underlying physiological mechanisms that may mediate these changes.

## Acknowledgments

I extend my sincere appreciation to the UN International Atomic Energy Agency (IAEA), Vienna for funding this work through regional project KEN 6025, within which this study was nested. I am grateful to the County Government of Nyeri for granting permission to conduct this study in their institution and to the administration of Kabarak University for allowing me to pursue my studies.

I deeply appreciate Dr. Peter Chege for his invaluable support in data analysis.

My heartfelt gratitude also goes to the staff at Nyeri Town Health Centre and the women of reproductive age who participated in this study—without their involvement, this research would not have been possible.

## Author contributions

**Conceptualization:** Purity Chepkorir Lang'at, Zipporah W. Ndung'u.

**Data curation:** Purity Chepkorir Lang'at, Dorcus Mbithe David-Kigaru, Zipporah W. Ndung'u.

**Formal analysis:** Purity Chepkorir Lang'at.

**Investigation:** Zipporah W. Ndung'u.

**Methodology:** Purity Chepkorir Lang'at, Dorcus Mbithe David-Kigaru, Zipporah W. Ndung'u, Pamela Kimeto.

**Project administration:** Dorcus Mbithe David-Kigaru, Zipporah W. Ndung'u.

**Resources:** Dorcus Mbithe David-Kigaru, Zipporah W. Ndung'u.

**Supervision:** Dorcus Mbithe David-Kigaru, Zipporah W. Ndung'u, Pamela Kimeto.

**Validation:** Pamela Kimeto.

**Writing – original draft:** Purity Chepkorir Lang'at, Zipporah W. Ndung'u.

**Writing – review & editing:** Purity Chepkorir Lang'at, Dorcus Mbithe David-Kigaru, Zipporah W. Ndung'u, Pamela Kimeto.

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
