## [Decision Letter · Decision Letter 0]

9 Jun 2025

PONE-D-25-16297Hormonal Contraceptives and Body Composition by Use of Stable Isotope Dilution Techniques Among Women of Reproductive Age in Nyeri County, KenyaPLOS ONE

Dear Dr. Lang'at,

Thank you for submitting your manuscript to PLOS ONE. After careful consideration, we feel that it has merit but does not fully meet PLOS ONE’s publication criteria as it currently stands. Therefore, we invite you to submit a revised version of the manuscript that addresses the points raised during the review process.

We look forward to receiving your revised manuscript.

Kind regards,

Ozan Karadeniz

Academic Editor

PLOS ONE

Reviewers' comments:

Reviewer's Responses to Questions

**Comments to the Author**

1. Is the manuscript technically sound, and do the data support the conclusions?

Reviewer #1: Partly

Reviewer #2: Yes

Reviewer #3: Yes

Reviewer #4: Yes

2. Has the statistical analysis been performed appropriately and rigorously? 

Reviewer #1: No

Reviewer #2: Yes

Reviewer #3: Yes

Reviewer #4: I Don't Know

3. Have the authors made all data underlying the findings in their manuscript fully available?

Reviewer #1: Yes

Reviewer #2: Yes

Reviewer #3: Yes

Reviewer #4: Yes

4. Is the manuscript presented in an intelligible fashion and written in standard English?

Reviewer #1: Yes

Reviewer #2: Yes

Reviewer #3: Yes

Reviewer #4: Yes

5. Review Comments to the Author

Reviewer #1: The objective of the paper is relevant, grouping all hormonal and non-hormonal contraceptives in two different categories may obscure important differences between them, particularly in how each type may contribute to changes in body composition. It would be helpful to clearly specify and justify why the different types were combined into a single category.

Reviewer #2: This manuscript presents an important and timely study on the association between hormonal contraceptive use and body composition among women in Nyeri County, Kenya, using stable isotope dilution techniques. The topic is relevant and the methodology is scientifically sound.

However, please confirm that this work has not been published or submitted elsewhere to address any concerns about dual publication.

I am certain that this paper could be a valuable contribution to the field.

Reviewer #3: Overall, this is a good study; however, some improvements are needed. The authors provided all necessary details regarding ethical approval and funding.

- Can the author justify why in the abstract, the sample size is reported as 114, while in the methods section, it is stated as 119, and at the endline, it is 104. Additionally, a 3-month follow-up is mentioned in the abstract, but the assessment methodology refers to a 6-month follow-up.

Introduction:

- Overall, the introduction is good but too brief.

- It would be beneficial to elaborate more, such as providing an overview of non-hormonal contraceptives, including their worldwide prevalence and specific statistics for Kenya, would be helpful.

- - If available, data/ percentage of women using hormonal contraceptive in Kenya, as well as the most commonly used methods worldwide and within Kenya.

- - It would enhance the study’s purpose to elaborate on the selection of "Kamakwa Ward," linking body composition to non-communicable diseases, such as diabetes. Are cardiovascular diseases also involved? This connection would strengthen the public health rationale for the study.

- More detail about “Stable Isotope Dilution Techniques (SIDT)” and which studies have employed these methods would be beneficial.

- Mentioning the reproductive age of the women studied would help in conveying the results to a broader, non-scientific audience.

- Clarification is needed on what the authors mean by “modern contraceptive” and what constitutes an “un-modern contraceptive.”

Methods, Study Design, and Population:

- It is important to specify which data collection tool was used for greater precision.

Statistical Analysis:

- The authors mention that multivariate regression analysis was conducted; however, no figures are provided to illustrate this regression.

- Including figures or graphs would be more helpful for the reader to compare than solely using tables.

Discussion:

- The first two paragraphs of the discussion repeat the results. It would be more effective if the authors compared their findings to those of international studies, highlighting where they are similar or diverge.

Referencing:

- Ensure that the references are formatted uniquely; for instance, in the introduction, the first reference is listed as (United Nations, 2019), while subsequent references should follow a consistent format.

- Reconsider the referencing in the discussion. For example, “...misinformation about contraceptive methods. Qualitative studies from Kenya (19, 20) and other parts of Africa (23) highlight that concerns about side effects are not represented in the literature; similarly, references 11, 12, and 15 were cited, while references 29 was cited before 28, and 30 and 31 were not cited at all.”

Reviewer #4: This article is well-structured and easily comprehensible. It provides valuable insights into the effects of hormonal contraceptives, a topic that often raises questions within the community. The project contributes to the scientific understanding of this issue and offers practical recommendations. The suggestion to provide dietary and nutritional advice to users of hormonal contraceptives is particularly useful. Such knowledge is crucial for healthcare workers, as the community frequently receives information from them.

Minor Corrections Needed:

Some minor corrections are needed regarding spelling and formatting.

Examples:

Reproductive Health and Contraceptive Use

The study shows THAT that 68.3% of the WRA were using some form of contraception, a figure

that compares favourably with national trends in Kenya, where the modern contraceptive

prevalence rate (mCPR) among married women is around 58% (17).

Delimitation of the Study

This study FOCUSSED on examining effects of hormonal contraceptives on the body composition

and nutritional status of women of reproductive age residing in Kamakwa Ward of Nyeri Central

Sub-County, Nyeri County Kenya.

6. PLOS authors have the option to publish the peer review history of their article (what does this mean? ). If published, this will include your full peer review and any attached files.

**Do you want your identity to be public for this peer review?** For information about this choice, including consent withdrawal, please see our Privacy Policy .

Reviewer #1: **Yes: ** Maria Cabrera Escobar

Reviewer #2: **Yes: ** Dr Ezechukwu Ikenna Nwokoma

Reviewer #3: No

Reviewer #4: No

---

## [Author Response · Author response to Decision Letter 1]

19 Jul 2025

RESPONSES TO REVIEWER’S COMMENTS

Manuscript Title: Hormonal Contraceptives and Body Composition by Use of Stable Isotope Dilution Techniques Among Women of Reproductive Age in Nyeri County, Kenya

We thank the editors and reviewers for their thorough and constructive comments. Below is our point-by-point response to each comment, with explanations of how the manuscript was revised. All page numbers refer to the revised manuscript.

Reviewer #1:

Comment: Grouping all hormonal and non-hormonal contraceptives into broad categories may obscure differences. Specify and justify grouping.

Response: Thank you for the observation. We have now clearly listed the specific types of hormonal (DMPA [injectables], oral pills, implants) and non-hormonal (copper IUDs, condoms) contraceptives used in our Methods section (Page 2 and 5). Justification for grouping is also provided due to sample size limitations and common mechanisms of action. This is also acknowledged in the Limitations section (Page 20).

Reviewer #2:

Comment: Confirm that the work has not been submitted or published elsewhere.

Response: We confirm that this manuscript has not been submitted or published elsewhere.

Reviewer #3:

Comment: Clarify inconsistencies in sample size (114 vs 119 vs 104); also, abstract mentions 3-month follow-up but methods describe 6 months.

Response: This has been corrected. The abstract, methods, and results sections now consistently state that 119 participants were recruited, 114 completed baseline SIDT, and 104 completed the endline at six months (Page 2 and 4). A 3-month mid-point check was conducted for compliance only; this has been clarified (Page 5).

Comment: Expand Introduction; provide prevalence data, rationale for setting, links to NCDs.

Response: We have expanded the Introduction to include contraceptive use prevalence globally and in Kenya, described types of non-hormonal methods, and linked contraceptive use to obesity and NCDs. We have also justified the selection of Kamakwa Ward based on urbanisation trends and NCD burden (Page 5).

Comment: Clarify use and validation of data collection tools.

Response: The Methods (Page 5) section has been updated to include: Use of standardized tools such as the Global Physical Activity Questionnaire (GPAQ) and the Women’s Dietary Diversity Score (WDDS). Details on pilot testing for cultural relevance and translation validation in the local setting (Page 5).

Comment: Include results of multivariate regression analysis and consider visual summary.

Response: We have now included the results of a multivariable logistic regression analysis that assessed the association between contraceptive use, physical activity level, dietary diversity score, and increased body fat percentage among women of reproductive age in Nyeri County, Kenya. As presented in Table 6 (Page 14)

Comment: Discussion repeats results; compare findings with existing literature.

Response: The Discussion (Pages 15–18) has been restructured to reduce repetition. Comparative analysis has been included using: Studies from Ethiopia (Mengesha et al., 2020), United States (Berenson et al., 2009), Uganda and South Africa (NAMSAL trial data, 2021), showing similar patterns in weight gain and body composition among hormonal contraceptive users.

Comment: Address referencing inconsistencies.

Response: Referencing format has been revised throughout the document for consistency and correct citation order.

Reviewer #4:

Comment: Minor language, grammar, and formatting issues.

Response: All noted typos and formatting issues have been corrected, and the manuscript has been revised for consistent use of British English.

We believe these changes have significantly strengthened the manuscript and are grateful for the opportunity to revise and resubmit.

Sincerely,

Signed

Purity Lang’at

---

## [Decision Letter · Decision Letter 1]

29 Aug 2025

Hormonal contraceptives and body composition by use of stable isotope dilution techniques among women of reproductive age in Nyeri County, Kenya

PONE-D-25-16297R1

Dear Dr. Chepkorir Lang'at, 

We’re pleased to inform you that your manuscript has been judged scientifically suitable for publication and will be formally accepted for publication once it meets all outstanding technical requirements.

Within one week, you’ll receive an e-mail detailing the required amendments. When these have been addressed, you’ll receive a formal acceptance letter, and your manuscript will be scheduled for publication.

An invoice will be generated when your article is formally accepted. Please note, if your institution has a publishing partnership with PLOS and your article meets the relevant criteria, all or part of your publication costs will be covered. Please make sure your user information is up-to-date by logging into Editorial Manager at Editorial Manager®  and clicking the 'Update My Information' link at the top of the page. For questions related to billing, please contact billing support .

If your institution or institutions have a press office, please notify them about your upcoming paper to help maximize its impact. If they’ll be preparing press materials, please inform our press team as soon as possible—no later than 48 hours after receiving the formal acceptance. Your manuscript will remain under strict press embargo until 2 pm Eastern Time on the date of publication. For more information, please contact onepress@plos.org.

Kind regards,

Ozan Karadeniz

Academic Editor

PLOS ONE

Reviewers' comments:

Reviewer's Responses to Questions

**Comments to the Author**

1. If the authors have adequately addressed your comments raised in a previous round of review and you feel that this manuscript is now acceptable for publication, you may indicate that here to bypass the “Comments to the Author” section, enter your conflict of interest statement in the “Confidential to Editor” section, and submit your "Accept" recommendation.

Reviewer #1: All comments have been addressed

Reviewer #3: All comments have been addressed

Reviewer #4: All comments have been addressed

2. Is the manuscript technically sound, and do the data support the conclusions?

Reviewer #1: Yes

Reviewer #3: Yes

Reviewer #4: Yes

3. Has the statistical analysis been performed appropriately and rigorously? 

Reviewer #1: Yes

Reviewer #3: Yes

4. Have the authors made all data underlying the findings in their manuscript fully available?

Reviewer #1: Yes

Reviewer #3: Yes

Reviewer #4: Yes

5. Is the manuscript presented in an intelligible fashion and written in standard English?

Reviewer #1: Yes

Reviewer #3: Yes

Reviewer #4: Yes

6. Review Comments to the Author

Reviewer #1: The authors have addressed most of my comments. I have provided minor additional comments in two documents. It appears that the document attached for revision contained multiple versions of the same file. I initially reviewed the original version, where my comments had not yet been addressed; only after completing that review did I notice a revised version further down, which I also reviewed. In hindsight, I should not have reviewed the earlier version. This situation was unusual for me and resulted in unnecessary time and some confusion.

Reviewer #3: The authors did great work in addressing the reviewers' comments. They expanded the introduction by adding valuable information, explained the use of various group and included the multiregression analysis, as well as edited the references section.

7. PLOS authors have the option to publish the peer review history of their article (what does this mean? ). If published, this will include your full peer review and any attached files.

**Do you want your identity to be public for this peer review?** For information about this choice, including consent withdrawal, please see our Privacy Policy .

Reviewer #1: No

Reviewer #3: No

Reviewer #4: No

---

## [Editor Report · Acceptance letter]

PONE-D-25-16297R1

PLOS ONE

Dear Dr. Lang'at,

I'm pleased to inform you that your manuscript has been deemed suitable for publication in PLOS ONE. Congratulations! Your manuscript is now being handed over to our production team.

Kind regards,

on behalf of

MD Ozan Karadeniz

Academic Editor

PLOS ONE